# Pembrolizumab-Induced Simultaneous and Refractory Systemic Capillary Leak and Cytokine Release Syndromes: A Case Report

**DOI:** 10.3390/curroncol32080469

**Published:** 2025-08-18

**Authors:** Eugénie Roberge-Maltais, Eric Lévesque, Vincent Castonguay, Nicolas Marcoux, Louis-Philippe Grenier, Martin Veilleux

**Affiliations:** 1Department of Internal Medicine, Université Laval, Quebec City, QC G1V 0A6, Canada; eugenie.roberge-maltais.med@ssss.gouv.qc.ca; 2Department of Hemato-Oncology, CHU de Quebec, Université Laval, Quebec City, QC G1V 4G2, Canada; eric.levesque.med@ssss.gouv.qc.ca (E.L.); vincent.castonguay.med@ssss.gouv.qc.ca (V.C.); nicolas.marcoux.med@ssss.gouv.qc.ca (N.M.); 3Department of Pharmacy, CHU de Quebec—Pavillon Hôtel-Dieu de Québec, Université Laval, Quebec City, QC G1R 2J6, Canada; louis-philippe.grenier@chudequebec.ca; 4Department of Internal Medicine, CHU de Quebec—Pavillon Hôtel-Dieu de Québec, Université Laval, Quebec City, QC G1R 2J6, Canada

**Keywords:** capillary leak syndrome, cytokine release syndrome, immune checkpoint inhibitors, PD-1 inhibitors, ruxolitinib, axitinib

## Abstract

This case report tells the story of a 40-year-old woman with advanced cervical cancer who developed two rare and serious complications after receiving immunotherapy (a treatment that helps the immune system fight cancer). She experienced severe body swelling, low blood pressure, and problems with multiple organs. Her condition was caused by an extreme immune reaction triggered by medication. Standard treatments did not work, but her doctors used a combination of advanced therapies that helped her recover after more than four months in the hospital. This case shows that while immunotherapy can be very effective against cancer, it may sometimes cause dangerous side effects—and highlights a new way of treating such complications.

## 1. Introduction

In recent years, the use of immune checkpoint inhibitors (ICIs), PD-1 inhibitors such as pembrolizumab, to treat malignancies has drastically improved the prognosis of several cancer types such as cervical cancer [1]. These agents increase the activity of the immune system by blocking the PD-1 protein which usually serves as a tumor survival mechanism by limiting T-cell mediated attacks on cancer cells [2]. By their intrinsic capability of decreased immune inhibition, ICIs are associated with a wide range of immune-related adverse events [3], some being organ-limited, and others being generalized inflammatory reactions. Cases of ICI-induced Systemic Capillary Leak Syndrome (SCLS) and of ICI-induced Cytokine Release Syndrome (CRS) have both been described [4,5].

SCLS is characterised by recurrent episodes of plasma extravasation and a classic triad of hemoconcentration, hypoalbuminemia and hypotension in absence of another cause of shock. Idiopathic SCLS was initially described in 1960 by Clarkson B, Thompson D, Horwith M et al. in their article Cyclical oedema and shock due to increased capillary permeability [6]. However, multiple secondary causes are now recognized, including ICI-induced [7,8]. A 2023 systemic review by Wong So and colleagues demonstrated that ICIs were associated with a significant over-reporting of SCLS compared with all other drugs [9].

CRS is a systemic inflammatory disease in which there is a massive release of cytokines [2]. Its clinical manifestations can range from mild (e.g., fever, malaise, myalgia) to life-threatening (e.g., shock, multi-organ failure) [10,11]. Inflammatory markers are elevated and coagulopathy, elevated liver enzymes and elevated creatinine are common findings [10,12]. A 2020 analysis of the World Health Organisation global pharmacovigilance database by Ceschi and colleagues identified 58 cases of ICI-induced CRS, of which 33 were associated with PD1 inhibitors [7].

We present here a rare case of a patient who was diagnosed with both pembrolizumab-induced SCLS and CRS in which we used a multimodal therapeutic approach tailored to the most probable underlying physiopathological mechanisms.

## 2. Case Report

A 40-year-old woman was admitted to the oncology department with anasarca and hypotension. Her relevant medical history included a metastatic squamous cell carcinoma of the cervix with a positive PD-L1 combined positivity score (CPS). The patient received treatment with six cycles of platinum (three cycles of cisplatin and three cycles of carboplatin), paclitaxel, bevacizumab and pembrolizumab. She thereafter began maintenance with both bevacizumab and pembrolizumab for six three-weekly cycles. She presented pembrolizumab-induced thyroid dysfunction after cycle four that was managed with levothyroxine but otherwise had no manifestation of immune toxicity on treatment. The patient had a radiologic complete response with this regimen.

Two weeks prior to admission, she had been hospitalized and investigated for anasarca. An echocardiogram revealed a 35–40% left ventricular ejection fraction with diffuse hypokinesis. Troponin level was not elevated, and a myocardial perfusion imaging (MIBI scan) showed absence of coronary artery disease (CAD). No evidence of neoplastic progression was found on PET scan. Cytologic analysis of ascites and pleural effusion were both negative. The decreased LVEF was attributed to probable bevacizumab-induced cardiomyopathy. However, the cause of her anasarca remained unclear but was attributed to possible heart failure. Before leaving the hospital, she received a dose of pembrolizumab alone and bevacizumab was discontinued permanently.

Three days after discharge, she presented again with worsening anasarca and hypotension. Upon initial evaluation, she had moderate ascites and bilateral pleural effusion. Her systolic blood pressure fluctuated between 70 and 100 mm of Hg. She was afebrile and pulse oximetry was 95% breathing ambient air. She weighed 71.9 kg and her normal weight was around 60 kg. Her initial biochemical parameters showed high hemoglobin of 173 g/L and hematocrit of 0.524. Serum albumin was low at 21 g/L and creatinine was raised to 154 μmol/L. Transaminase levels were elevated, with an ALT of 299 U/L. Initial troponin was normal, and C reactive protein was elevated at 40.1 mg/L (Table 1).

She was initially diagnosed with pembrolizumab induced serositis, for which she was treated with corticosteroids. However, her condition did not improve. Alternative causes of anasarca were excluded by exhaustive investigation. There was no evidence of cirrhosis or of portal vein thrombosis on an abdominal CT scan. A full screening for hepatopathies was negative. No evidence of nephrotic syndrome was found on urinalysis and her renal function normalised during hospitalisation. Though she had a reduced LVEF, it was deemed improbable that this was the cause of her anasarca as it did not explain her low albumin and hemoconcentration. Furthermore, she did not respond well to diuretic therapy despite furosemide 20 mg IV four times daily and amiloride 10 mg twice a day. No alternative causes of hypotension and dyspnea, such as sepsis or massive pulmonary embolism, were identified. Thus, pembrolizumab-induced SCLS emerged as the most probable cause of her symptoms. The result of the Naranjo Adverse Drug Reaction Probability Scale was 7. A summary of the pointage is depicted in Appendix A.

During hospitalisation, the patient developed multiple organ dysfunction. While she presented with elevated transaminases, no cause of hepatic dysfunction was found on abdominal imaging or in a complete hepatopathy work up. She was thus diagnosed with pembrolizumab-induced auto-immune hepatitis. A few days after admission, she developed fluctuating confusion and drowsiness. Multiple electroencephalograms showed signs of encephalopathy and seizures. A brain MRI showed non-specific bilateral anomalies of the corona radiata. No signs of inflammation or infection were found on a lumbar puncture. An auto-immune encephalitis panel was negative. Her symptoms did not resolve with lactulose administration. Neurologists diagnosed her with pembrolizumab-induced auto-immune encephalitis with associated seizures. Finally, ten days after admission, the patient’s troponin I and NTproBNP values markedly increased up to 12,001 ng/L and 10,275 ng/L, respectively. On an echocardiogram, her LVEF had lowered to 25–30% with an apical ballooning pattern. As coronary arterial disease had previously been ruled out, two possible diagnoses remained: stress-induced or pembrolizumab-induced auto-immune cardiomyopathy. Concurrently, the patient developed disseminated intravascular coagulation. While her C-reactive protein, which was initially elevated, rapidly decreased under corticosteroid therapy, a cytokine panel showed diffusely high cytokine levels Table 2. These multiple immune-mediated organ dysfunctions combined with diffuse elevated cytokines, despite a high dose of corticosteroid, supported the diagnosis of pembrolizumab-induced CRS.

With both SCLS and CRS diagnoses confirmed, literature was reviewed, and recommended treatments were attempted. At first, high-dose corticosteroids were trialed and increased up to 1 g of methylprednisolone daily. As her anasarca worsened and she developed encephalitis and cardiomyopathy under corticosteroids alone, intravenous immunoglobulins (IVIG) were added as they seemed to be one of the most effective treatments for SCLS according to a systematic review by Tae Seong and colleagues [13]. She responded well; her neurological symptoms and anasarca improved. After introduction of IVIGs, corticosteroids were tapered. Plasmapheresis were briefly attempted, but they were not tolerated. A series of immunosuppressant medications were also trialed. Mycophenolate mofetil and Rituximab both failed to control the underlying auto-immune process. Upon receiving the results of the cytokine panel (Table 2), ruxolitinib was introduced as it acts on many cytokines through the JAK/STAT pathway [14] and has been used to successfully treat refractory CRS cases associated with CAR-T cell therapy [15]. A summary of all treatments received is depicted in Appendix A. The patient’s auto-immune manifestations markedly improved. For her SCLS, theophylline was introduced as literature suggested methylxanthines were effective [8], but it was discontinued because of toxicity. Finally, a tyrosine kinase inhibitor, axitinib, was initiated as it was used successfully in ICI-induced SCLS in case reports [4]. Concurrently, the patient received diuretics and albumin as needed to maintain adequate volemia. Octreotide was later added as it had been reported to help control ascites [4]. A summary of biochemicals parameters and evolution of patient’s weight during hospitalization is shown in Appendix A and Figure 1.

After over four months of hospitalisation, she was finally discharged. Her hemoglobin was 74 g/L, her albumin was 22 g/L, and her ALT was 126 U/L. She weighed 54.7 kg. Her treatment regimen included weekly IVIG infusions and axitinib 3 mg twice daily for her SCLS. Lanreotide 30 mg monthly, weekly albumin infusions and diuretics were continued to maintain adequate volemia. Her CRS was successfully being treated with ruxolitinib 20 mg twice daily. She remained stable at home, requiring only minor adjustments to her diuretics. Unfortunately, recurrent squamous cell carcinoma was identified on ascites fluid and pleural effusion cytology two months after discharge. Her cancer progressed significantly to the point where she could no longer take oral medication. Her anasarca reappeared suddenly and massively after stopping her axitinib. Unfortunately, her health rapidly deteriorated. It was decided with her family to pursue palliative care 9 months after initial signs of SCLS.

## 3. Discussion

In this case report, we present an unusual presentation of a patient who developed pembrolizumab-induced SCLS and CRS simultaneously and was then treated with a multimodal therapeutic approach. Indeed, SCLS was treated with a combination of IVIGs, axitinib and diuretics while the CRS was treated with ruxolitinib after corticosteroids failed to control the disease.

The treatments used to control the patient’s SCLS were based on previously published experience with similar clinical settings [4,13]. Indeed, a systematic review demonstrated that IVIGs were the most effective treatment for idiopathic SCLS, surpassing methylxanthines and corticosteroids [13]. We found IVIGs to be effective in her case as her symptoms drastically improved with these recurrent infusions. An osmotic effect was mentioned as possible mechanism, but no amelioration was noted after albumin infusion. Corticosteroids and theophylline were attempted but had little if any effect. Axitinib was used in combination with IVIGs by Haixia and colleagues to successfully treat a case of pembrolizumab-induced SCLS [4]. Our case report corroborates this finding as axitinib improved our patients’ symptoms enough for her to require less frequent treatment with IVIGs and lead to a successful discharge attempt from the hospital. The symptoms worsened upon discontinuation of axitinib at time of disease progression, also supporting its role in controlling SCLS.

The physiopathological mechanisms underlying SCLS are unclear. However, literature suggests that adhesion molecules and vascular growth factors might be elevated and involved in causing the vascular leak [9,13]. The Vascular Endothelial Growth Factor (VEGF) pathway is involved in angiogenesis and regulation of vascular permeability [16]. It is highly likely that axitinib is effective in SCLS by blocking VEGF receptors involved in one of the downstream pathways causing fluid extravasation (Appendix A) [17]. Bevacizumab is itself an anti-VEGF agent and represents a potential treatment option for capillary leak syndrome secondary to immunotherapy. It is noteworthy that discontinuation of bevacizumab may have contributed to the decompensation of the immunotherapy-induced capillary leak syndrome. Of course, other alternative causes such as nephrotic syndrome, cardiac dysfunction and cirrhosis were ruled out.

CRS is an uncontrolled inflammatory reaction resulting from a massive release of cytokines [18]. Its manifestations are often non-specific, but life-threatening cardiac or neurologic toxicities are frequent, as was the case with our patient [18]. We did a cytokine panel to see which inflammatory pathways were specifically involved. It showed that her cytokine levels were diffusely elevated. As her symptoms were refractory to corticosteroids, we began treatment with ruxolitinib to act on multiple cytokines at once by blocking the JAK/STAT pathway (Appendix A). It was effective in controlling her systemic inflammation. This strategy had previously been used in CRS cases associated with CAR-T cell therapy [15]. Cases of ICI-induced CRS have been reported in literature. However, they have mostly used corticosteroids and tocilizumab, which act on IL-6 [5,11]. In the context of our patients’ diffusely elevated cytokine levels, we believed tocilizumab would not have been the best treatment option. Also, there are studies and publications that indirectly compare or evaluate the effectiveness of tocilizumab and ruxolitinib in the treatment of cytokine release syndrome (CRS) secondary to immunotherapy, particularly CAR-T cell therapy and immune checkpoint inhibitors (ICIs). However, direct comparative studies (head-to-head randomized clinical trials) remain limited.

To our knowledge, this is the first case report of a patient developing both ICI-induced SCLS and CRS. The use of a JAK inhibitor to treat pembrolizumab-induced CRS has not yet been reported. It is a promising treatment avenue as it acts on multiple cytokines that may be elevated in CRS, as was demonstrated by our patient’s cytokine panel. The main limitation of this report is that multiple medications were attempted before the combination of IVIGs, axitinib and ruxolitinib. Therefore, we cannot rule out that the efficacy observed was not partially due to those previous treatments.

## 4. Conclusions

In conclusion, we presented here a unique case of a patient who simultaneously developed pembrolizumab-induced refractory SCLS and CRS. While she showed no improvement on corticosteroids, treatment with a combination of IVIGs and axitinib for her SCLS and ruxolitinib for her CRS appeared to alleviate her symptoms. This novel treatment approach utilises multiple action mechanisms tailored to the complex physiopathology that appears to be involved in both diseases. Further reports and studies are needed to assess the efficacy of this drug combination in refractory systemic pembrolizumab-induced inflammatory cases and the role of cytokine panels to further individualize care for these complex complications.

## Figures and Tables

**Figure 1 curroncol-32-00469-f001:**
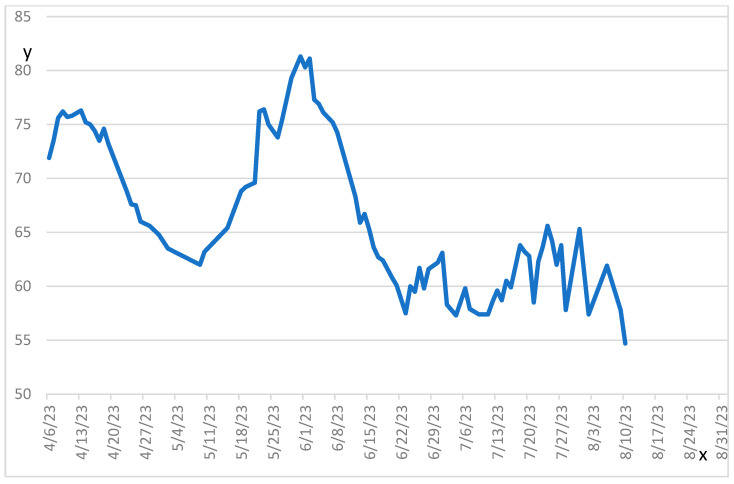
Evolution of the patient’s weight throughout hospitalisation.

**Table 1 curroncol-32-00469-t001:** Evolution of the patient’s biochemical parameters.

Date	Hb (g/L)	Albumin (g/L)	Creatinine	ALT (U/L)	Troponin	Weight (kg)
Day 1	179	21	154	299	12	N/A
Day 3	159	N/A	105	269	N/A	71.9
Day 4	158	19	109	244	N/A	73.5
Day 11	170	20	100	98	12,001	75.2
Day 25	117	25	48	231	512	65.6
Day 45	93	26	63	272	328	68.8
Day 57	85	25	39	172	N/A	80.3
Day 70	89	33	48	120	N/A	66.7
Day 86	77	34	34	46	N/A	62.2
Day 110	77	26	49	20	N/A	58.7
Day 127	95	29	57	44	N/A	65.3
Day 147	76	26	48	90	N/A	54.7
Day 151	78	24	52	115	N/A	N/A
Day 165	83	26	64	132	N/A	N/A
Day 195	97	28	79	62	N/A	N/A

N/A: Not available.

**Table 2 curroncol-32-00469-t002:** Cytokine panel (despite use of steroids).

Cytokine	Value (pg/mL)	Reference Value (pg/mL)
Group A: Innate autoimmune inflammation		
IL-1 alpha	59.7	0–58.3
IFN-alpha2	<2.5	5–134
IL-17E/IL-25	1925	37–1194
Group B2: T helper cell mediated inflammation		
IL-12p70	57.7	0–14.6
Group B3: Innate inflammation/Cytokine storm		
IL-6	12.9	0.2–10.2
IL-18	4.6	7–109
IP-10	472	15–220
M-CSF	196	2–192
Group D: Type 2/Type 3 immune response		
IL-28A	483	0–272
TGF alpha	32.5	1–31.2
Group F: Hematopoietic growth factors		
IL-7	29.3	0–20.5
Group G: Homeostatic chemokines		
CTAK	1859	292–1685
Group H: Platelet activation/wound healing		
PDGF-AA	5015	173–3619
PDGF-AB/BB	>37,500	6653–34,584
All other cytokine levels were within normal range		

## Data Availability

Data are contained within the article.

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
