# Peer review of "Pembrolizumab-Induced Simultaneous and Refractory Systemic Capillary Leak and Cytokine Release Syndromes: A Case Report"

_curroncol, 2025, doi:10.3390/curroncol32080469_

Round 1

Reviewer 1 Report

Comments and Suggestions for Authors

The manuscript is well written and organized with a potential novel treatment strategy for ICI induced toxicity:

  1. I think the conclusion is overstated as this is a case report and multiple studies need to done to give this conclusion
  2. Supplementary tables are missing which were mentioned in the text and treatment summary.
  3. Although, tocilizumab has been established for treatment of CRS, there is no comparison between tocilizumab and ruxolitinib in treatment of CRS in the discussion area or introduction. 
  4. It would be more beneficial to illustrate more on why tocilizumab was not used and why it was not appropriate for the case.
  5. Including more recent citations would strengthen the manuscript's relevance and academic rigor.
  6. Figure 1: x & y axis needs to be labelled 
  7. Line 136: misspelling of "biochemichal" 
  8. More information is needed in the discussion area about axitinib might mitigate SCLS
  9. Although pembrolizumab can be the main cause of these symptoms but there are other confounding factors that need to be addressed or clearly state how did the authors came to with this conclusion and eliminate the other confounding factors, such as bevacizumab-induced vascular toxicity was ruled out. It might be a cumulative toxicity not from pembrolizumab alone

Author Response

Here is the response to reviewer attached. 

Reviewer 2 Report

Comments and Suggestions for Authors

The authors have presented a relevant and insightful case report that makes a meaningful contribution to the understanding of clinical manifestations resulting from severe immune-related adverse events (irAEs) induced by Programmed Cell Death Protein 1 (PD-1) inhibitors, specifically pembrolizumab, in the treatment of advanced cervical cancer. This report effectively illustrates the potential dangers of immune checkpoint blockade therapy and proposes a novel approach to managing such complications. The findings described herein hold considerable value for clinicians, oncologists, and researchers in refining diagnostic protocols, therapeutic planning, and clinical decision-making strategies.

However, several modifications and clarifications are necessary to enhance the scientific rigor and clarity of the manuscript:

  1. In line 26: The statement “after elimination of alternative causes” requires further elaboration. The authors are encouraged to enumerate and describe the alternative diagnoses that were considered and excluded.
  2. In lines 39–40: Repetition of the abbreviation PD-1 is unnecessary, as it was already defined in lines 22–23. The redundancy should be avoided to maintain coherence.
  3. In lines 40–41, the authors should cite appropriate references to support the statements made regarding mechanisms or outcomes related to pembrolizumab-induced adverse effects.
  4. Did the authors utilize the Naranjo Adverse Drug Reaction Probability Scale to assess the causality of the drug reaction? If so, this analysis should be explicitly included and summarized in the manuscript to strengthen the evidence linking pembrolizumab to the adverse events described.
  5. Did the authors monitor any electrolyte abnormalities during the analysis? If such data exist, the authors are encouraged to present them in a tabular format and provide a discussion on the clinical significance of electrolyte fluctuations during PD-1 inhibitor-induced immune reactions in the patient.
  6. In lines 203–205, the pathway explanation provided is unclear. The authors should consider elaborating this section, potentially by including a diagrammatic representation of the signaling mechanisms involved—particularly those related to the JAK/STAT pathway—to enhance conceptual clarity.

Author Response

Here is the response to the reviewer attached.
